# FLOOD SIMULATION WITH PHYSICS-INFORMED MESSAGE PASSING

## ABSTRACT

Flood modeling is an important tool for supporting preventive and emergency measures to mitigate flood risks. Recently, there has been an increasing interest in exploring machine learning-based models as an alternative to traditional hydrodynamic models for flood simulation to address challenges such as scalability and accuracy. However, current ML approaches are ineffective at modeling the early stages of flooding events, limiting their ability to simulate the entire evolution of the flood. Another key challenge is incorporating physics domain knowledge into these data-driven models. In this paper, we address these challenges by introducing a physics-inspired graph neural network for flood simulation. Given a (geographical) region and precipitation data, our model predicts water depths in an autoregressive fashion. We propose a message-passing framework inspired by the conservation of momentum and mass expressed in the shallow-water equations, which describe the physical process of a flooding event. Empirical results on a dataset covering 9 regions and 7 historical precipitation events demonstrate that our model outperforms the best baseline, and is able to capture the propagation of water flow better, especially at the very early stage of the flooding event.

## 1 INTRODUCTION

Flooding is the natural hazard with the greatest social and economic impact in the United States and affects lives and livelihoods around the world (Tellman et al., 2021; Eckstein et al., 2021; National Academies of Sciences, Engineering, and Medicine; Jha et al.; PBL Netherlands Environmental Assessment Agency). In cities, flooding generates direct property damage, indirect losses through supply chain disruption, and threats to livelihood through drowning and interruption of transportation networks, limiting access to healthcare and emergency services (Haraguchi & Lall; Han et al.; Gori et al.; Panakkal et al., a;b). With climate change increasing the intensity and frequency of extreme precipitation in many regions (on Climate Change , IPCC), urbanization reducing natural flood protection (Merz et al., 2014; Sebastian et al., 2019), and rapid population growth in flood-prone regions, the severity of urban flooding is projected to continue to grow (Berkhahn et al., 2019; Schreider et al., 2000).

Many mechanisms can drive flooding. Here, we focus on flooding in urban areas, driven by both fluvial (rising river levels) and pluvial (intense rainfall) mechanisms. The evolution of flood depth in urban areas is a time-evolving physical process typically represented by the shallow-water equations, a set of partial differential equations (PDEs) that describe a thin layer of fluid of constant density in hydrostatic balance, bounded from below by the bottom topography and from above by a free surface. In practice, flooding in urban areas is modeled using specialized solvers, such as LISFFOOD-FP (Shaw et al., 2021) and HEC-RAS (Brunner, 2016), which not only solve the relevant PDEs but also account for space- and time-varying rainfall, evolving inundation regions, and additional features. A key insight of the past decade is that increasing the resolution of models is often a better use of limited computational resources than including detailed representations of physical processes (Bates; Bates et al.). However, these models remain computationally expensive, require extensive calibration with poor generalizability, and perform poorly on small and medium-scale floods for which processes like drainage infrastructure and infiltration, neglected by most models, are important (Rosenzweig et al.; Saksena & Merwade). This limits their utility for vitally important applications including real-time flood warning, probabilistic hazard assessment, representation of green infrastructure benefits, and optimization of infrastructure design.

Recently, machine learning (ML) methods have emerged as a promising alternative to hydrodynamic numerical models due to their flexibility, efficiency, and scalability (Mosavi et al., 2018; Bentivoglio et al., 2022). ML-based models for flooding have generally fallen into three groups. The first uses time series models such as Long-Short Term Memory (LSTM) networks to predict the time series of discharge (flow/time) at a single location (Wi & Steinschneider, 2022; 2023; Nevo et al., 2021), trained on gauge observations. While this approach has proven flexible and skillful in transfer learning tasks, it does capture the spatially varying dynamics of urban flooding. The second approach predicts the maximum extent of a flood, given information on the area affected and the storm, trained on high water marks and satellite observations (Muñoz et al., 2021; Berkhahn et al., 2019; Kabir et al., 2020; Löwe et al., 2021; Hofmann & Schüttrumpf, 2021). However, such approaches do not provide information on the time evolution of the system, critical for many applications, and may be difficult to check for physical realism. The third approach, like 2D hydrodynamic models, considers both spatial and temporal dynamics of flooding. The primary limitation of this approach is that observations are not, in general, available, so models are trained in "surrogate" mode on the output of computationally expensive models (Bates, 2022). This approach is the focus of our paper.

In this work, we propose ComGNN, a physics-inspired graph neural network for flood simulation. GNNs have achieved promising results in predicting physics simulations (Pfaff et al., 2021; Lino et al., 2022), including fluid dynamics problems Keisler (2022); Lam et al. (2022). They support a wide range of PDE discretizations, such as regular and irregular meshes (Brandstetter et al., 2022). Our proposed model takes as input a directed graph derived from the flow direction of a region where each mesh cell is a node with an outgoing edge to its steepest neighbor cell. At each time step, each node is first considered as an isolated bucket that accumulates its current water volume and water from the rain, which is later propagated to the surrounding nodes using a message-passing inspired by the conservation of momentum and mass. These features enable our method to simulate the early stages of flooding events much better than current approaches. Our work makes the following contributions to emerging field of GNN-based flood simulation:

- We propose ComGNN, a graph neural network for urban flood dynamics given spatially and temporally varying rainfall that operates in a two-stage paradigm: (1) retain water where it falls and (2) propagate water to surroundings.

- We propose a message-passing mechanism on the flow direction graph that is explicitly designed based on the conservation of momentum and mass for water propagation.

- We evaluate our method on 9 watersheds (regions) for 7 historical floods each. Our experiments show that our approach outperforms current approaches, with up to one order of magnitude lower RMSE in the early stages of a flooding event.

## 2 RELATED WORK

**Machine learning for spatial and temporal variability of floods.** The focus in ML for flood prediction has mainly been on modeling either the spatial or temporal variability of floods. For instance, ML has been used to predict water flow over time at a single location (Wi & Steinschneider, 2022; 2023; Nevo et al., 2021). There have been applications of ML to the prediction of flood inundation, susceptibility, and hazard maps (Wang et al., 2020; Guo et al., 2022; Löwe et al., 2021; Oliveira Santos et al., 2023; Farahmand et al., 2023). Mosavi et al. (2018) provide a comprehensive review of machine learning approaches for flood prediction. Bentivoglio et al. (2022) review machine learning applications to flood mappings. However, the interplay between predictions of spatial and temporal variabilities is extremely important for ML to be used as an alternative to current 2D hydrodynamic models for flood simulation. There have recently been a few works addressing this problem. For instance, Kazadi et al. (2022) proposes a GNN that predicts water depth and velocities as vector features in an auto-regressive manner. However, this work does not account for rainfall precipitations and fails to simulate floods from very early stages (when the domain is dry). Bentivoglio et al. (2023) also proposes a GNN architecture inspired by the shallow-water equations. The key advantages of our method is that (i) it takes spatially distributed rainfall data as input; (ii) it explicitly emulates the conservation of momentum and mass of the shallow-water equations; (iii) it operates in two stages to propagate water over a region, which is crucial for early stage of flood simulations; (iv) it is trained directly on real-world data instead of synthetic data.

**Machine learning for the simulation of dynamical systems.** More recently, Graph Neural Networks (GNNs) have been successfully applied to physics-based simulations (Sanchez-Gonzalez et al., 2019; Kipf et al., 2018; Fortunato et al., 2022; Cranmer et al., 2020; Battaglia et al., 2016; Allen et al., 2022). Due to their ability to handle irregular graph data, they support different PDE discretization schemes (e.g, structured and unstructured meshes). GNN message passing and graph Laplacian operator have been associated with differential operators, suitable for solving PDEs (Brandstetter et al., 2022; Maddix et al., 2022). For instance, Pfaff et al. (2021) propose a GNN that can simulate the dynamics of fluids, rigid solids, and deformable materials. Alet et al. (2019) learns a continuous-space function representation with a learnable discretization of the domain into finite elements. Two GNNs for weather forecasting were recently shown to achieve promising results by Lam et al. (2022) and Keisler (2022). Another line of recent works that have improved the learning of PDE dynamics are neural operators (Lu et al., 2021; Anandkumar et al., 2019; Patel et al., 2021; Li et al., 2021; Yin et al., 2023). These methods do not require any knowledge about the underlying PDEs that describe the physical process. They operate on continuous space and time by taking as input an initial condition state and predicting the continuous state at a given time. Neural operators have also shown promising results in weather forecasting (Pathak et al., 2022). Flood simulation, however, poses new challenges to existing ML approaches for dynamical systems due to (i) the nature of real-world datasets, which cover large and have complex topographies and (ii) the need for accounting for external precipitation data. Our work addresses these issues and experimental results show that our approach outperforms the ones proposed by Pfaff et al. (2021) and Brandstetter et al. (2022) in the context of flood simulation.

## 3 FLOOD MODELING: MATHEMATICAL FRAMEWORK

The theoretical framework for flood modeling is based on fluid mechanics described by the 3D Navier-Stokes equation. In practice, however, the characteristic vertical length scale of the flow is very small with respect to the characteristic horizontal length scale, resulting in a constant horizontal velocity field throughout the depth of the fluid. The dynamics of a flooding process are, therefore, derived by depth integrating the 3D Navier-Stokes equation, leading to a system of non-linear PDEs called shallow-water equations (de Almeida et al., 2012). The shallow water equations, without convective acceleration and negligible friction, are defined as follows.

$$\frac{\partial h}{\partial t} + \nabla \cdot \mathbf{q} = 0 \qquad \text{(conversation of mass)} \tag{1}$$

$$\frac{\partial \mathbf{q}}{\partial t} + gh\nabla(h + z) = \mathbf{0} \qquad \text{(conversation of momentum)} \tag{2}$$

where $h(x, y; t)$ is the water depth relative to the ground elevation $z(x, y)$, $\mathbf{q} = (q_x(t), q_y(t))$ is the discharge (per unit width), $\nabla = (\frac{\partial}{\partial x}, \frac{\partial}{\partial y})$ is the spatial gradient operator.

## 4 PROBLEM FORMULATION AND APPROACH

### 4.1 PROBLEM FORMULATION

Given a region $\mathcal{R}$, represented as a graph, and a spatially distributed rainfall event $P^{1:K}$ over $K$ time steps, our goal is to simulate the early subsequent unfolding states (water depth levels) $\mathcal{H}^{1:K}$ of $\mathcal{R}$ due to $P^{1:K}$ over a region defined by its bare ground topographic surface.

### 4.2 METHOD

We propose ComGNN, a novel GNN for flood simulation based on the retention and dispersion of water. In the water **retention phase**, we consider each cell $v_i$ as an isolated bucket with no water exchange with its surrounding cells. The water level in $v_i$ is represented by the latent features $\mathbf{e}_i^t(\in \mathbb{R}^d)$, solely depending on the rainfall $p_i^t(\geq 0)$ and previous water level $h_i^{t-1}(\geq 0)$.

$$\mathbf{e}_i^t = \text{MLP}([p_i^t \| h_i^{t-1}]) \tag{3}$$

where MLP is a multi-layer perceptron, and $\|$ is the concatenation operation.

The **dispersion phase** acts as a learning-based spatial solver of the shallow-water equations. Following the method of lines (Schiesser, 2012), we first define a scheme for the spatial domain. Based on the Taylor expansion, the $n^{th}$-derivative of $f$ of order $n$ can be approximated as

$$\frac{\partial^n f(x)}{\partial x} = \sum_{i=1}^{N} \alpha_i f(y_i) \tag{4}$$

where $\alpha_i$ are coefficients, and $y_i$ are points sampled in the neighborhood of $x$, that is, $y_i = x + \Delta x_i$. For instance, first-order forward finite difference can be recovered from Eq 4 by setting $N = 2$, $\Delta x_1 = 0, \alpha_1 = -\frac{1}{\Delta x_2}$, and $\alpha_2 = \frac{1}{\Delta x_2}$ (See Appendix A.1). When the coefficients $\alpha_i$ are learnable from the sampled points, it gives rise to an adaptive approximation scheme with different orders of accuracy for each point $x$. Furthermore, Eq 4 can be seen as a special case of the more general message passing $\psi$ and node update operation $\phi$ in GNNs (Brandstetter et al., 2022).

$$\frac{\partial^n f}{\partial x}(x) = \phi\left(\{\psi(f(y_i), f(x))\}_{y_i \in \mathcal{N}(x)}\right) \tag{5}$$

where $\mathcal{N}(x)$ is the neighborhood of $x$. By setting $\psi$ to a scaling factor of its first argument $f(y_i)$, and $\phi$ to the summation of its arguments, we recover Eq 4. Applying the spatial derivative from Eq 5 to the conservation of momentum (Eq. 2) at each node $i$ gives:

$$\frac{\partial \mathbf{q}_i}{\partial t} + g\phi\left(\{\psi(h_j, z_j, h_i, z_i)\}_{v_j \in \mathcal{N}_{\mathbf{out}}(i)}\right) = \mathbf{0} \tag{6}$$

where $\mathcal{N}_{\mathbf{out}}(i) = \{v_j | v_i \rightarrow v_j\}$. We define the message passing $\psi$ as a backward difference to model the ability of flow going from $v_i$ to $v_j$.

$$\psi(h_j, z_j, h_i, z_i) = \sigma(\mathbf{e}_i) \odot \mathrm{MLP}((\mathbf{e}_i + \mathbf{z}_i) - (\mathbf{e}_j + \mathbf{z}_j)) \tag{7}$$

where $\sigma$ is the sigmoid function, $\odot$ is the element-wise multiplication, $\mathbf{z}_{i/j} = \mathrm{MLP}(z_{i/j})$, and $\mathbf{e}_{i/j}$ is the latent representation of the water retained from the rain (Eq. 3) which we substitute for the water depth $h_{i/j}$. There is flow from $v_i$ to $v_j$ if there is a difference in water surface, that is, $(\mathbf{e}_i + \mathbf{z}_i) - (\mathbf{e}_j + \mathbf{z}_j)$. Furthermore, this flow can only happen if there is water in $v_i$, hence, multiplication by $\sigma(\mathbf{e}_i)$ as a gating mechanism. By defining the node update $\phi$ as a summation of its arguments and applying forward Euler time integrator to Eq 6, we have

$$\mathbf{q}_i^t = \mathbf{q}_i^{t-1} + \Delta t g \sum_{v_j \in \mathcal{N}_{\mathbf{out}}(i)} \sigma(\mathbf{e}_i^t) \odot \mathrm{MLP}((\mathbf{e}_i^t + \mathbf{z}_i) - (\mathbf{e}_j^t + \mathbf{z}_j)) \tag{8}$$

where $\mathbf{q}_i^t$ can be regarded as the total flow going out of $v_i$.

After obtaining $\mathbf{q}_i^t$ at each cell $v_i$, the second level message passing of the water dispersion phase is computed with the following (implicit) time integration of the equation of the conservation of mass.

$$h_i^t = h_i^{t-1} + \Delta t \phi\left(\{\psi(\mathbf{q}_i^t, \mathbf{q}_j^t)\}_{v_j \in \mathcal{N}_{\mathbf{in}}(i)}\right) \tag{9}$$

where $\mathcal{N}_{\mathbf{in}}(i) = \{v_j | v_j \rightarrow v_i\}$. By setting $\psi$ as an identity function and $\phi$ as a parametric function (MLP) of the summation of incoming flows from $v_j$ minus outgoing flow of $v_i$, we obtain

$$h_i^t = h_i^{t-1} + \Delta t \, \mathrm{MLP}(\sum_{j \in \mathcal{N}_{\mathbf{in}}(i)} \mathbf{q}_j^t - \mathbf{q}_i^t) \tag{10}$$

Multiple iterations of this bi-level message-passing (Eq. 8 & 10) can be performed to simulate dispersion over long distances. Intermediate states can be interpreted as latent space forecasting (Migus et al., 2023); $h_i^t$ will be the output of the final message-passing—this is analogous to a multistep time integration of $h_i^t$. At the next time step $t + 1$, $p_i^{t+1}$ and $h_i^t$ are fed back into our model, in an auto-regressive manner, for the prediction of $h_i^{t+1}$. The entire process is illustrated in Figure 1.

$\Delta t$ and the gravitational force are fixed, only appear as constant multiplicative factors, we assume them to be equal to 1 in (Eq. 8 & 10) in the present work.

**Region representation as a graph.** In practice, region surfaces are represented in a raster format (digital elevation model –DEM), where each pixel/grid cell represents the ground elevation. The first challenge in developing a GNN for flooding simulation is to design a graph topology that

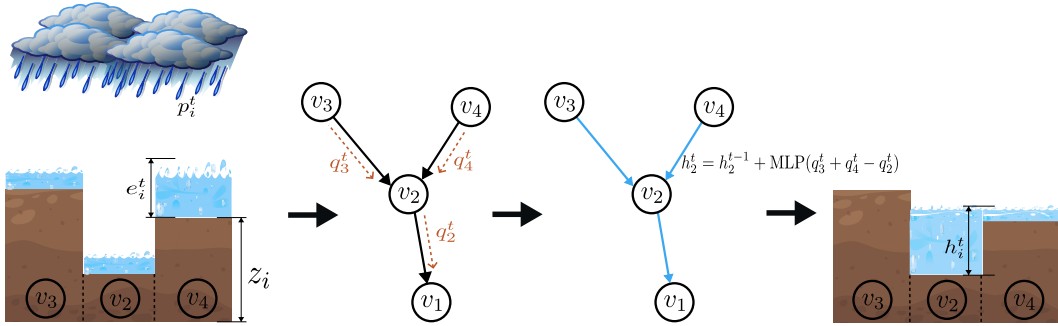

Figure 1: **Water retention and dispersion**. Each cell in the domain representing a region is considered as an isolated bucket filled with water $e_i^t$ from rain $p_i^t$. Water is propagated by computing, with possibly many iterations, discharges $q_i^t$ (by conserving momentum Eq. 8), and getting the more stable water depth $h_i^t$ (by conserving mass Eq 10).

captures the dynamics of the flooding process, especially at an early stage of the process when water tends to flow from high elevations to lower ones. We convert a given region $R$ into a directed graph $G_R(V, E)$, which remains static during the entire simulation. $G_R$ is defined as the D8 flow direction map (Jenson & Domingue, 1988) based on the DEM of $R$ (See Appendix A.3). Each grid cell $v_i (\in V)$ is considered as a node, and a single directed outgoing edge $e_{i \to j} (\in E)$ connects $v_i$ to its steepest neighbor $v_j$. Each cell $v_i$ has as features a rainfall time series $p_i^{1:K}$ (corresponding to its location in $P^{1:K}$), initial state $h_i^0$, and ground elevation $z_i$. Based on this configuration Eq 8 can further be reduced to the following:

$$\mathbf{q}_i^t = \mathbf{q}_i^{t-1} + \sigma(\mathbf{e}_i^t) \odot \text{MLP}((\mathbf{e}_i^t + \mathbf{z}_i) - (\mathbf{e}_j^t + \mathbf{z}_j))$$

**Loss function**. At each time $t$, we propose the following loss function as the objective to minimize the discrepancy between predicted water levels $h_i^t$ and the ground truth $w_i^t$:

$$\mathcal{L}_{\text{total}} = \mathcal{L}_{\text{diff}} + \mathcal{L}_+ \tag{11}$$

$$\mathcal{L}_{\text{diff}} = \sum_i^N \begin{cases} |h_i^t - w_i^t| & , |h_i^t - w_i^t| < 1 \\ (h_i^t - w_i^t)^2 & , \text{otherwise} \end{cases} \tag{12}$$

$$\mathcal{L}_+ = \frac{1}{N} \sum_i^N \max(0, -h_i^t) \tag{13}$$

where $N$ is the number of nodes. The loss $\mathcal{L}_{\text{diff}}$ is a combination of the $L_1$ loss (for very small values) and $L_2$ (for larger values). $\mathcal{L}_+$ penalizes negative values of $h_i^t$ returned by the model. In our experiments, $\mathcal{L}_{\text{diff}}$ performs better than $L_1$ or $L_2$ used individually. This can be explained by the fact that we are dealing with very sparse data, and the optimization can adapt to different regimes of the learning process when the loss is small (by switching to $L_1$) or large (by switching to $L_2$) without manually tuning the learning rate.

## 5 EXPERIMENTS

We evaluate our approach for flood simulation (ComGNN) using a representative dataset of flooding events and evaluation metrics. We hypothesize that ComGNN is a more accurate surrogate for hydrodynamic models than existing alternatives from the literature. Due to space limitations, we provide more details on datasets, experimental settings, and additional results in the Appendix.

### 5.1 DATASET

Experiments are based on the simulations from the hydrodynamic model LISFLOOD-FP, version 8 (Shaw et al., 2021). We consider 9 sub-watshed regions from Harris County in Texas. For each of these regions, simulations were run using 7 historical rainfall events (based on the flood history in

Harris County[1] collected from NOAA NEXRAD radar precipitation records from the Multi-Radar Multi-Sensor Gauge Corrected (MRMS-GC) Quantitative Precipitation Estimation (QPE) product (Martinaitis et al., 2020). These are hourly recorded precipitation values, which we reduce to 5-minute intervals using a linear interpolation between consecutive time steps (see Appendix A.4).

The Harris County, TX, area, which includes the city of Houston (the 4th most populous in the United States), is the ideal case study for the evaluation of flood simulation methods. The region has experienced multiple severe floods in the past decades and is investing billions of dollars in flood mitigation (HCFCD, 2019) and faces broad climate adaptation challenges representative of those facing urban watersheds across the U.S. (ASFPM, 2020).

**Data normalization**  We normalize the Digital Elevation Model (DEM) of each region independently using standardization. This helps in handling situations where regions have similar topographies but different altitudes as they are expected to show similar inundation behaviors. Since the precipitation and water depth are highly sparse, we log-transform them using $\log(1 + \frac{x}{1e-2})$ Pathak et al. (2022), followed by a division by 10.

## 5.2 Experimental Settings

**Baselines**  We compare our model (ComGNN) to the following approaches: (i) **U-net** (Ronneberger et al., 2015), arguably the most popular CNN model for predicting dynamic systems defined on a mesh; (ii) **ConvLSTM** (SHI et al., 2015); (iii) **MP-PDE** (Brandstetter et al., 2022), a GNN solver for PDEs; and (iv) **MeshGraphNet** (Pfaff et al., 2021), proposed for the simulation of dynamic systems. To assess the efficacy of our proposed message-passing, we replace it with a **GCN** (Kipf & Welling, 2017) and a **GAT** (Veličković et al., 2018) keeping other parts of the model architecture the same. We further consider a variant of our model (**ComGNN⁻**) where the precipitation data is directly used as a node feature, instead of operating in two stages (i.e., retention and then propagation). This model is similar to the one proposed by Bentivoglio et al. (2023) and it will serve as a subject in the ablation analysis. We also compare our model to simple methods such as a multi-layer perceptron (**MLP**), a model that predicts the previous depth at each node (**Persitence**), and another that simply computes the next water depth by incrementing it with the amount of rain (**Rain-Incr**). The purpose of these simple methods is to show that when the initial state of the region has already water in it, they can perform better than some of the most sophisticated methods. Similar to our model (ComGNN), all baseline architectures are applied in an auto-regressive manner.

**Training setup**  We conducted a thorough hyperparameter search on both our model and baselines and selected the configuration with the lowest RMSE score on the validation set (See Appendix A.5 for more details). The dataset has a total of 63 combinations of (watershed) regions and precipitation data, from which 9 were used as the training set, 3 as the validation set, and the remaining ones as our test set. The simulation lead time was set to 40, the largest we could train on a single NVIDIA GPU Ampere A40. For each sample in the validation set, we trained one model instance, resulting in an ensemble of 3 models per method.

**Evaluation metrics**  We use the root mean square error (RMSE), the Nash–Sutcliffe model efficiency coefficient (NSE), the Pearson correlation coefficient (r), and the critical success index (CSI) for performance evaluation (See Appendix A.2).

## 5.3 Results

Our evaluation will focus on the accuracy of the water depths predicted by ComGNN for multiple flood simulations. We will compare our approach against a comprehensive set of baselines to demonstrate that the proposed physics-inspired message-passing architecture is able to approximate complex of flooding events. Two important aspects that will be considered in our evaluation are the the prediction lead time and the initial stage of the flooding event. Longer lead times are more challenging to be predicted by autoregressive models due to the accumulation of errors over time. Moreover, we will show that the early stages of the flood—when the soil is still dry—are harder to be predicted by existing machine learning models.

---

[1]https://www.hcfcd.org/About/Harris-Countys-Flooding-History

Table 1: Accuracy/error of simulations over all of the test watershed regions and precipitation events combined. The simulation is run over 40 time steps using true water depth values at time $t = 0$ as initial input. We use $t_r$ to denote time relative to the beginning of the simulation. For CSI, we only show results at $t_r = 40$ for thresholds $\gamma = \{0.001, 0.01\}$. The results show that our approach achieves the best results across all evaluation metrics.

| Method | RMSE ↓ | | NSE ↑ | | $r$ ↑ | | CSI ↑ ($t_r = 40$) | |
|---|---|---|---|---|---|---|---|---|
| | $t_r = 20$ | $t_r = 40$ | $t_r = 20$ | $t_r = 40$ | $t_r = 20$ | $t_r = 40$ | $0.001m$ | $0.01m$ |
| Persistence | .3052 | .8559 | .4663 | .4407 | .0000 | .0000 | .0000 | .0000 |
| Rain-Incr | .1972 | .5465 | .6766 | .6590 | .7253 | .7013 | .7122 | .4653 |
| ConvLSTM | .1780 | .4397 | .7199 | .7491 | .7823 | .8181 | .7051 | .4691 |
| MLP | .2119 | .5457 | .6445 | .6597 | .6807 | .7156 | .5306 | .4404 |
| GCN | .1874 | .5482 | .6986 | .6576 | .7859 | .7669 | .7014 | .4993 |
| GAT | .2155 | .6103 | .6366 | .6078 | .7020 | .7042 | .6949 | .3286 |
| U-net | .2329 | .4546 | .6001 | .7364 | .7488 | .8022 | .6581 | .4734 |
| MeshGraphNet | .1597 | .4807 | .7615 | .7141 | .8327 | .7968 | .6120 | .5412 |
| MP-PDE | .1824 | .5192 | .7098 | .6817 | .7936 | .7895 | .7158 | .5209 |
| ComGNN$^-$ | .1571 | .4830 | .7674 | .7121 | .8412 | .7782 | .6180 | .5637 |
| ComGNN | **.1328** | **.3615** | **.8218** | **.8154** | **.8866** | **.8854** | **.7463** | **.6486** |

Tables 1 and 2 show results when the true water depths at times $t = 0$, and $t = 40$ are used as initial conditions, respectively. The purpose is to show how models perform at different stages of a flooding event. In Table 1, corresponding to the very early stage of a flooding event when the land is completely dry, we can see that all baseline methods struggle to capture the dynamics of the water flows, which is evidence that current approaches are ineffective at handling low water depth values. Our proposed model ComGNN shows the best performance across different metrics, performing even better than its variant ComGNN$^-$, which does not consider water retention. the decrease in performance of MP-PDE and MeshGraphNet can be explained by the fact the domain in a flooding scenario involves complex topography defined by the ground elevation, whereas these models were mainly designed for systems on simpler manifolds, such as lines, and 2D planes.

Table 2: Similar results to those shown in Table1 but using true water depths at time $t = 40$ as initial input. We note that at a later stage of the flood—when there is already water in the domain—the baselines perform much better, including simple ones such as Rain-Incr. Our approach (ComGNN) also achieves competitive results in this setting.

| Method | RMSE ↓ | | NSE ↑ | | $r$ ↑ | | CSI ↑ ($t_r = 40$) | |
|---|---|---|---|---|---|---|---|---|
| | $t_r = 20$ | $t_r = 40$ | $t_r = 20$ | $t_r = 40$ | $t_r = 20$ | $t_r = 40$ | $0.001m$ | $0.01m$ |
| Persistence | 1.0343 | 1.8319 | .6250 | .4747 | .7676 | .5708 | .5115 | .3395 |
| Rain-Incr | .6407 | .9765 | .8129 | .7608 | .8838 | .8304 | .8395 | .7338 |
| ConvLSTM | **.4213** | **.6668** | **.9095** | **.8721** | **.9500** | **.9238** | .8352 | .8129 |
| MLP | .6663 | .9722 | .8006 | .7624 | .8962 | .8707 | .7677 | .7250 |
| GCN | .7469 | 1.1933 | .7617 | .6805 | .8818 | .8310 | .8428 | .7575 |
| GAT | .7871 | 1.2652 | .7422 | .6545 | .8763 | .8301 | .8467 | .7053 |
| U-net | .6704 | .8794 | .7987 | .7968 | .8669 | .8746 | .7997 | .7108 |
| MeshGraphNet | .7327 | 1.1963 | .7686 | .6794 | .8778 | .8048 | .8209 | .7897 |
| MP-PDE | .7429 | 1.2169 | .7636 | .6719 | .8799 | .8018 | .8606 | .7587 |
| ComGNN$^-$ | .6835 | 1.0015 | .7924 | .7515 | .8924 | .8390 | .8184 | .7887 |
| ComGNN | .5333 | .7686 | .8625 | .8370 | .9280 | .9034 | **.8666** | **.8284** |

Table 2 shows results when flood state at time $t = 40$ is used as our initial condition. At this stage, when there is already water in the domain, we can notice that most methods start to perform much better, including the naive method Rain-Incr, which is competitive with the most sophisticated methods. This shows that simulating the flow of water can become a trivial process when there is already enough water in the domain. Furthermore, this shows that existing methods are better suited for settings where there is already a significant amount of water in the domain but underperform when the

water is sparsely distributed. Nonetheless, we can also notice a decrease in the performance of our method, which can be explained by the fact that the flow direction graph used as the representation of a region might not be adequate to represent the actual flow when the level of water rises over an entire region—as the discharge is less dependent on elevation. See Table 4 in Appendix A.3 for a comparison between the flow direction graph and grid-based graph.

Since there might be different times at which there is water in a given region during a rainfall event, in Table 3, we show results when the initial condition for each domain is one time step before there is any flooded area. Here again, we can see that our proposed method outperforms other approaches.

Table 3: Similar results to those shown in Table Table1 but using as input the time immediately before there is any water in the region. Our approach (ComGNN) excels at this challenging setting, consistently outperforming the baselines.

| Method | RMSE $\downarrow$ | | NSE $\uparrow$ | | $r \uparrow$ | | CSI $\uparrow$ ($t_r = 40$) | |
|---|---|---|---|---|---|---|---|---|
| | $t_r = 20$ | $t_r = 40$ | $t_r = 20$ | $t_r = 40$ | $t_r = 20$ | $t_r = 40$ | $0.001m$ | $0.01m$ |
| Persistence | .4457 | 1.0485 | .4536 | .4237 | .0000 | .0000 | .0000 | .0000 |
| Rain-Incr | .2986 | .6470 | .6491 | .6588 | .6855 | .7099 | .7092 | .4862 |
| ConvLSTM | .2484 | .5164 | .7277 | .7519 | .7912 | .8215 | .7291 | .5506 |
| MLP | .3025 | .6458 | .6431 | .6596 | .6757 | .7288 | .6117 | .4637 |
| GCN | .2891 | .6653 | .6636 | .6462 | .7484 | .7703 | .7151 | .5140 |
| GAT | .3249 | .7379 | .6097 | .5975 | .6636 | .7107 | .7274 | .3864 |
| U-net | .2818 | .5333 | .6750 | .7398 | .7741 | .8055 | .7074 | .5223 |
| MeshGraphNet | .2458 | .5911 | .7319 | .6982 | .8021 | .7916 | .6784 | .5511 |
| MP-PDE | .2786 | .6283 | .6800 | .6719 | .7615 | .7881 | .7492 | .5197 |
| ComGNN$^-$ | .2450 | .5731 | .7331 | .7111 | .8093 | .7850 | .6896 | .5768 |
| ComGNN | **.2037** | **.4344** | **.7990** | **.8107** | **.8686** | **.8849** | **.7816** | **.6603** |

Under the same initial conditions as in Table 3 Figure 2 shows the absolute error of our model ComGNN (Figures 2b, 2e, 2h, and 2k) and the absolute error of best-performing baseline ConvL-STM, (Figures 2c, 2f, 2i, and 2l) over the region represented in Figure 2m at lead times 10, 20, 30, and 40. The first column (Figures 2a, 2d, 2g, 2j) represents the true flood map (water depths) states. We can see that ComGNN achieves lower errors than ConvLSTM. We provide further visualizations of the correlation between the true and predicted water depth values of all the methods in Appendix A.6 where we show that ComGNN's predictions are the most aligned with the true water depths.

## 6 Conclusion

We have presented ComGNN, a physics-inspired graph neural network for early-stage flood simulation based on a given rainfall event. ComGNN operates in two stages: at each time step, water from the rain is first stored in the area of the region where it falls (*water retention*), and it is then propagated to the surrounding areas (*water dispersion*) using a message-passing that mimics the conservation of momentum and mass. A region is represented as a directed graph by linking each cell/node to its steepest neighbor based on the D8 flow direction of the region's topography.

Our experiments were based on realistic simulations of 7 historical floods over 9 watershed regions in Harris County, Texas. Results have shown that ComGNN is effective at simulating the early stage of a flooding event, outperforming existing methods with significant margins in terms of different evaluation metrics (RMSE, NSE, Pearson's coefficient of correlation, and CSI).

As future work, we will address the degradation in the performance of ComGNN when the level of water rises. Since a D8 flow direction graph becomes ineffective in capturing the flow of very deep waters, we will investigate how to dynamically change the graph representation of a region based on the current water surface elevation (water depth + ground elevation) and potential energy surface.

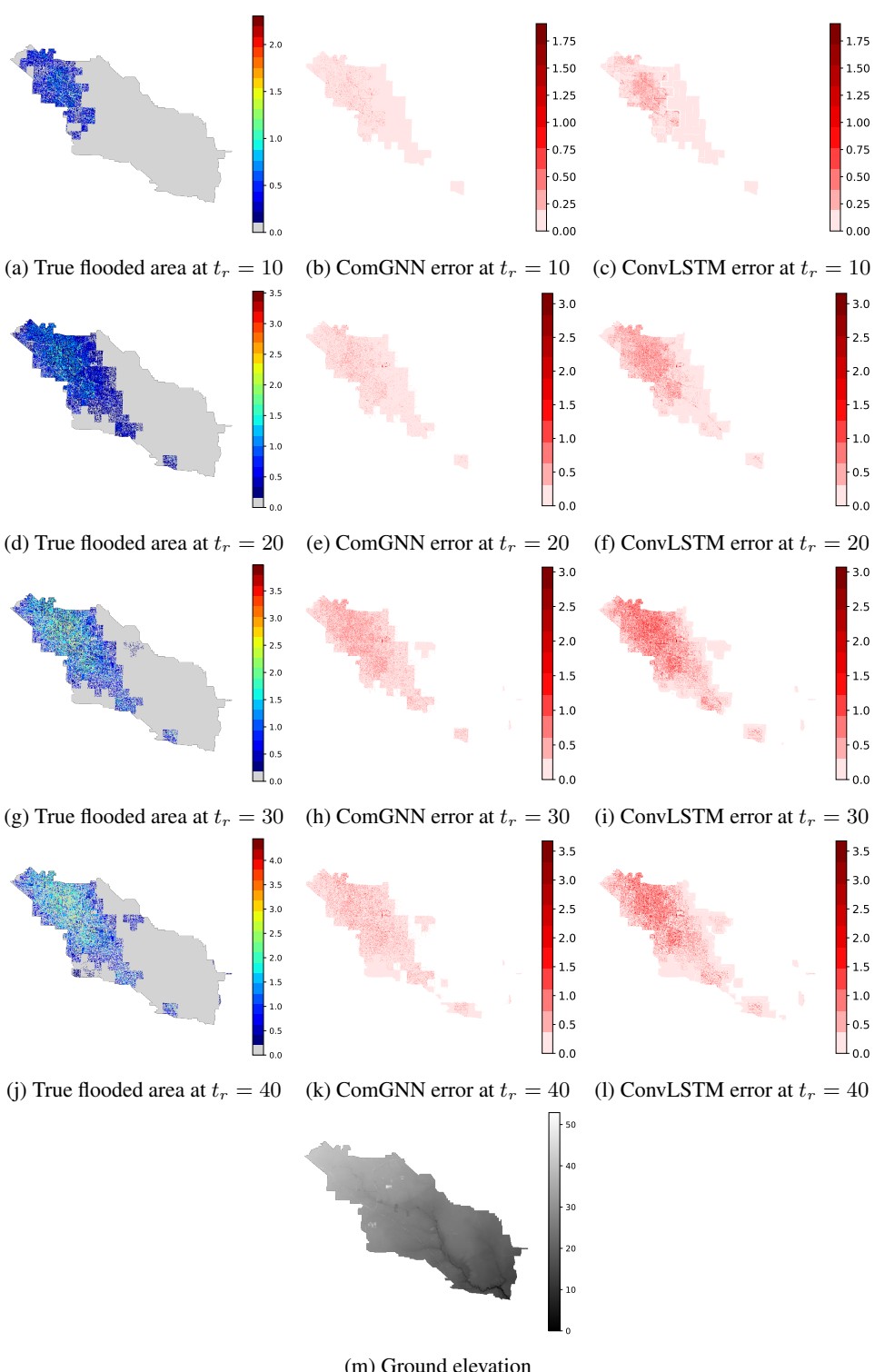

(a) True flooded area at $t_r = 10$   (b) ComGNN error at $t_r = 10$   (c) ConvLSTM error at $t_r = 10$

(d) True flooded area at $t_r = 20$   (e) ComGNN error at $t_r = 20$   (f) ConvLSTM error at $t_r = 20$

(g) True flooded area at $t_r = 30$   (h) ComGNN error at $t_r = 30$   (i) ConvLSTM error at $t_r = 30$

(j) True flooded area at $t_r = 40$   (k) ComGNN error at $t_r = 40$   (l) ConvLSTM error at $t_r = 40$

(m) Ground elevation

Figure 2: Absolute error of our proposed method (middle column) and the absolute error of Con-vLSTM (right column) compared to the true flood area (left column) at lead times 20 (row 1) and 40 (row 2) for the region represented in Figure2m. The results show that ComGNN achieves lower error than the baseline, which is consistent with the results from Table 1.

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

## A  APPENDIX

### A.1  DERIVATIVE APPROXIMATION WITH TAYLOR SERIES

For simplicity, let us assume a one-dimensional domain. Suppose we want to approximate the
derivate of a smooth enough function $f$ at point $x_i$ with $l$ points to the left, and $r$ points to the right,
forming a stencil that includes the points $x_j$ such that $i - l \leq j \leq i + r$. Let us further assume $x_j$
are uniformly, that is, $\Delta x_j = j \Delta x$. The Taylor expansion of $f$ at $x_j$ centered at $x_i$ is

$$f(x_j) = f(x_i) + \frac{j\Delta x}{1!} f_x(x_i) + \frac{(j\Delta x)^2}{2!} f_{xx}(x_i) + \frac{(j\Delta x)^3}{3!} f_{xxx}(x_i) + \frac{(j\Delta x)^4}{4!} f_{xxxx}(x_i) + \dots$$

where $i - l \leq j \leq i + r$. Multiplying each of these expansions by a constant $c_j$ and summing them
up gives

$$\sum_{j=i-l}^{i+r} c_j f(x_j) - \left( \sum_{j=i-l}^{i+r} c_j \right) f(x_i) = \left( \sum_{j=i-l}^{i+r} jc_j \right) \frac{\Delta x}{1!} f_x(x_i) + \left( \sum_{j=i-l}^{i+r} j^2 c_j \right) \frac{(\Delta x)^2}{2!} f_{xx}(x_i)$$
$$+ \left( \sum_{j=i-l}^{i+r} j^3 c_j \right) \frac{(\Delta x)^3}{3!} f_{xxx}(x_i)$$
$$+ \left( \sum_{j=i-l}^{i+r} j^4 c_j \right) \frac{(\Delta x)^4}{4!} f_{xxxx}(x_i)$$
$$+ \dots \tag{14}$$

Eq. 14 provides a way to approximate higher order derivatives at any order accuracy of $f$. For
instance, first-order derivative at third-order accuracy can be obtained by setting $\left( \sum_{j=i-l}^{i+r} jc_j \right)$ to 1
and $\left( \sum_{j=i-l}^{i+r} j^2 c_j \right)$ to 0.

### A.2  EVALUATION METRICS

At each time step $t$, we use the root mean square error (RMSE), the Nash–Sutcliffe model efficiency
coefficient (NSE), and the Pearson correlation coefficient ($r$) for performance evaluation:

$$\text{RMSE} = \sqrt{\frac{1}{N} |y_i - p_i|^2} \qquad \text{NSE} = 1 - \frac{\sum_i^N |y_i - p_i|_2^2}{\sum_i^N |y_i - \bar{y}_i|_2^2} \qquad r = \frac{\sum_i^N (y_i - \bar{y}_i)(p_i - \bar{p}_i)}{\sqrt{\sum_i^N (y_i - \bar{y}_i)^2 \sum_i^N (p_i - \bar{p}_i)^2}}$$

where $y_i$ is the true value and $p_i$ is the predicted value. We also consider the critical success index
(CSI) that measures the spatial accuracy of the classification of cells as flooded or non-flooded areas
for a given flooding threshold $\gamma$. CSI is evaluated as follows:

$$\text{CSI} = \frac{\text{TP}}{\text{TP} + \text{FP} + \text{FN}}$$

where TP are true positives (cells with both the predictions and ground truths greater than $\gamma$), FP
are false positives (cells whose ground truths are less than $\gamma$ but the model's predictions are greater
than $\gamma$), and FN are false negatives (cells where the model fail to predict a flooded area). In our
experiments, we consider $\gamma = \{0.001 \text{ m}, 0.01 \text{ m}\}$ since we are dealing with very shallow waters.

## A.3 D8 FLOW DIRECTION GRAPH

We used the tool ArcGIS Pro [2] to generate the D8 flow direction graph of a region based on its digital elevation model (DEM). D8 (eight-direction) indicates that the output direction of a cell is related to its 8 adjacent cells. The direction is coded as an unsigned 8-bit integer, with 1 denoting east, 2 south-east, 4 south, 8 south-west, 16 west, 32 north-west, 64 north, and 128 north-east. We generate a directed graph by considering a cell as a node, and by adding an outgoing edge to the adjacent cell corresponding to the direction code.

**Comparison between Flow Direction Graph and Grid-based Graph**    Since our proposed model ComGNN is based on the flow direction, we instead choose GCN to compare performances when a region is represented as a flow direction and a grid-based graph (when the mesh is directly used as the graph). In Table 4 the GCN with flow direction is denoted as plain GCN and the one with grid-based graph GCN-grid. We can that with flow direction graph, results are relatively good in early states, that is, with initial condition at t = 0. GCN-grid starts performing better when there is water in the domain (initial condition t=40) as at this stage, especially, when water evens out, the direction of the propagation becomes less important.

Table 4: Comparison between D8 flow direction graph and grid-based graph representation.

| Method | RMSE ↓ | | NSE ↑ | | $r$ ↑ | | CSI ↑ ($t_r = 40$) | |
| | $t_r = 20$ | $t_r = 40$ | $t_r = 20$ | $t_r = 40$ | $t_r = 20$ | $t_r = 40$ | 0.001m | 0.01m |
| --- | --- | --- | --- | --- | --- | --- | --- | --- |
| Initial condition $t = 0$ (see Table 1 | | | | | | | | |
| GCN | **.1874** | **.5482** | **.6986** | **.6576** | **.7859** | **.7669** | .7014 | **.4993** |
| GCN-grid | .2046 | .5702 | .6603 | .6397 | .7179 | .7282 | **.7081** | .4239 |
| Initial condition $t = 40$ ( see Table 2) | | | | | | | | |
| GCN | .7469 | 1.1933 | .7617 | .6805 | .8818 | .8310 | .8428 | **.7575** |
| GCN-grid | **.7252** | **1.1243** | **.7722** | **.7058** | **.8902** | **.8518** | **.8527** | .7564 |
| Initial condition same as in Table 3 | | | | | | | | |
| GCN | **.2891** | **.6653** | **.6636** | **.6462** | **.7484** | **.7703** | **.7151** | **.5140** |
| GCN-grid | .3084 | .6836 | .6342 | .6337 | .6843 | .7398 | .7138 | .4524 |

## A.4 DATASET

We consider 9 sub-watershed regions from Harris County in Texas, all shown together in Figure 3.

Figure 4 shows the ground elevations of the 9 watersheds. They are represented in a raster format where each pixel cell represents a $30m \times 30m$ area. The areas and dimensions (in terms of number of rows and columns) of the watersheds are given in Table 5. Details about the precipitation data are shown in Table 6.

**Simulations**    The flood data were generated using LISFLOOD-FP(Shaw et al., 2021), a two-dimensional hydrodynamic model specifically designed to simulate floodplain inundation over complex topography by numerically solving the shallow water equations. It predicts water depths in each cell of the discretized domain using an adaptive time stepping. We provided the DEM (ground elevation) of a region and prediction data as input and collected snapshots of water depth states as output every 5 minutes of the simulation process clock-time. It is worth noting that between output intervals, LISFLOOD-FP computes several internal time steps.

## A.5 MODEL CONFIGURATIONS

The learning rate was set to 1e-4 for all the models. We also noticed that all the models considered in our work performed much better with the loss function proposed in Eq. 11, with a bump in

---

[2]https://pro.arcgis.com/en/pro-app/latest/tool-reference/
spatial-analyst/how-flow-direction-works.htm

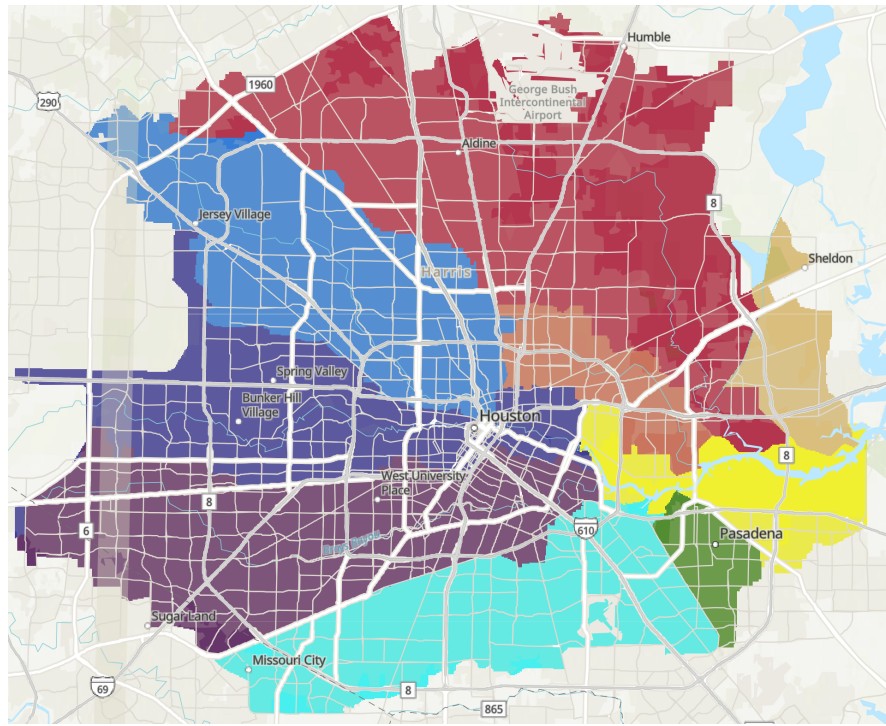

Figure 3: Color-coded watershed regions considered in our work

Table 5: Regions considered in this work with the areas and dimensions in terms of the number of rows and columns in their raster

| Region | Area (km$^2$) | Rows | Columns |
|---|---|---|---|
| White Oak Bayou | 288 | 1083 | 749 |
| Vince Bayou | 41 | 280 | 370 |
| Sims Bayou | 242 | 1412 | 562 |
| San Jacinto River | 272 | 745 | 406 |
| Hunting Bayou | 77 | 514 | 417 |
| Greens Bayou | 549 | 1512 | 1032 |
| Carpenters Bayou | 65 | 331 | 558 |
| Buffalo Bayou | 267 | 1360 | 750 |
| Brays Bayou | 330 | 1358 | 577 |

Table 6: Preciptiation data

| Rainfall Event | Date | Intensity (mm/s) | |
|---|---|---|---|
| | | mean | max |
| Pre-Memorial Day Flood | May 13, 2015 | 0.4 | 33.9 |
| Memorial Day Flood | May 25, 2015 | 2.6 | 97.7 |
| N/A | Oct 31, 2015 | 6.1 | 146.5 |
| Tax Day Flood | Apr 17, 2016 | 3.1 | 73.3 |
| Hurricane Harvey | Aug 25, 2017 | 5.6 | 122.4 |
| N/A | Jul 04, 2018 | 3.4 | 85.4 |
| Tropical Storm Imelda | Sep 17, 2019 | 2.7 | 103.3 |

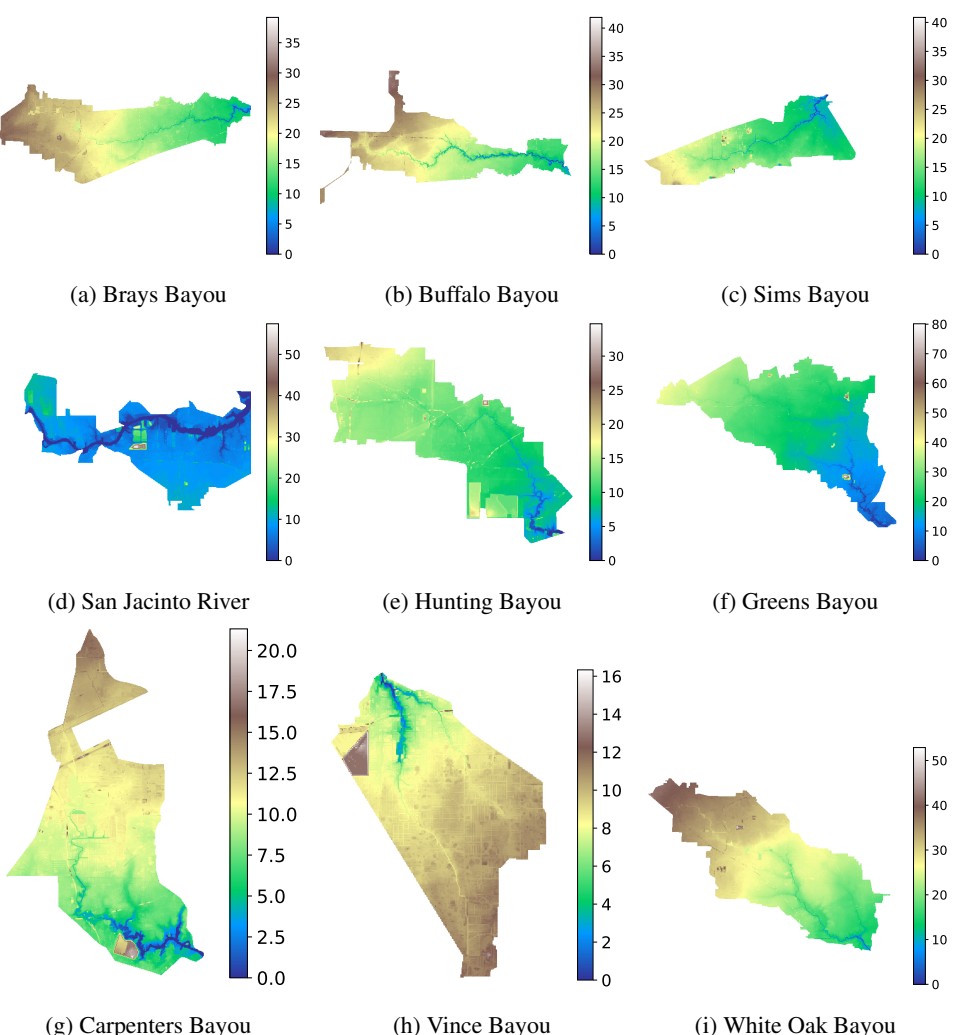

Figure 4: Absolute error of our proposed method (middle column) and the absolute error of U-net (right column) compared to the true flood area (left column) at lead times 10 (row 1), 20 (row 2), 30 (row 3), 40 (row 4) of the region represented in Figure2m.

performance of up to 30% in some cases. *tanh* function showed better performance compared to the original activation functions of some of the baselines. All linear transformations were used without the bias term, this seemed to help to deal with the sparsity of the data. The configurations with the best performance of individual methods are given below,

- **Persistence**: naive and non-parametric method that assumes that the conditions at the time of prediction will not change. That is, this method predicts the same values provided as the initial conditions for all subsequent predictions. This method seems to perform well if the conditions do not change much, which is usually the case when there is already water in the domain.

- **Rain-Incr**: simple method we implement as $hi^t = h^{t-1} + \alpha p^t$; where $h^t$ is the water depth at time $t$, $p^t$ is the precipitation at time $t$, and $\alpha$ is a learnable parameter. This method simply increments the current amount of water by the rain, which also seems to perform well in an area where there is already water in the domain.

- **ConvLSTM** (SHI et al., 2015): re-implementation with all CNN components with 64 channels and kernel of size 3.

- **MLP**: implementation with 3 layers with 32 neurons in each layer.

- **GAT/GCN** with 2 or 3 layers performed about the same. We kept 2 layers to reduce the number of parameters, and therefore avoid overfitting.

- **U-net** implementation with 2 down-samplings and 2 up-samplings all with 32 channels. The Swish activation function implementation from MP-PDE was used here as it increased performance.

- **MeshGraphNet**: re-implementation following description from the original paper Pfaff et al. (2021). One layer of the proposed method seemed to perform the best, with *tanh* as the activation. For this method in particular the loss function in Eq. 11, improved the performance by a significant margin compared to $L_2$ loss. Note that no spatial coordinates were used in this implementation version, given the huge sizes of the domains.

- **MP-PDE** (Brandstetter et al., 2022): Adapted from the original implementation. One layer performed the best, and the prediction was conducted for one step ahead to match the configurations of other approaches used in our work. Spatial coordinates are not used like in the original implementation either. The Swish activation function was left unchanged since it performed better than *tanh* and ReLU.

- **ComGNN** showed better performance with a 3-layer MLP for Eq. 3, one layer of Eq. 8, and 2 layers of Eq. 10). *tanh* was used as the activation function and all the layers were implemented with 32 neurons.

Table 7 shows the numbers of parameters of models considered in this work.

Table 7: Number of parameters of each model considered in this work

| Method | # parameters |
|---|---|
| Rain-Incr | 1 |
| ConvLSTM | 157,376 |
| MLP | 1,152 |
| GCN | 4,688 |
| GAT | 4,816 |
| U-net | 37,408 |
| MeshGraphNet | 10,832 |
| MP-PDE | 14,024 |
| ComGNN$^-$ | 12,944 |
| ComGNN | 12,928 |

### A.6 CORRELATION BETWEEN PREDICTIONS AND TRUE VALUES

We provide visualizations of the correlation between true and predicted water depth values in Figures 5, 6, 7 and 8 for lead times 10, 20, 30, and 40, respectively. The results demonstrate that ComGNN's predictions are the most aligned with the true water depth values.

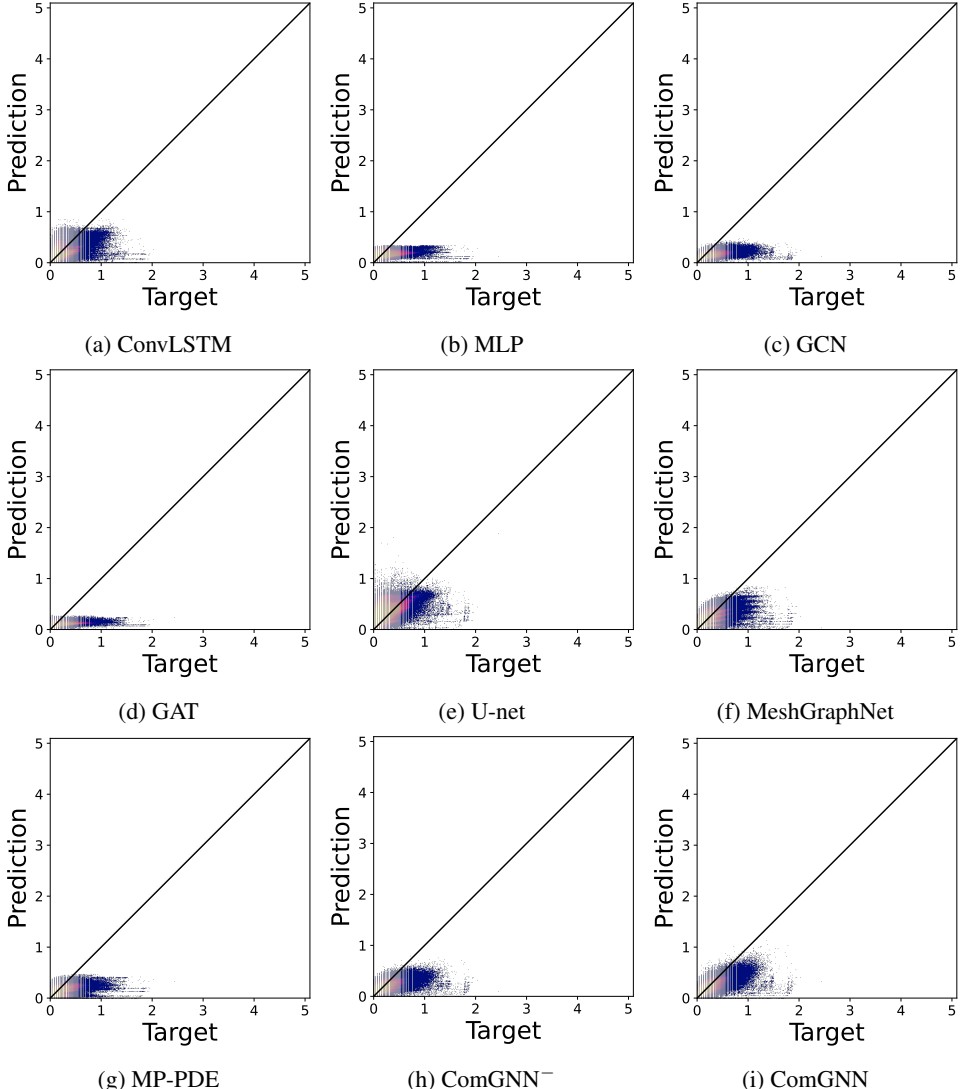

Figure 5: Scatter Plots at lead time 10 in log-log scale

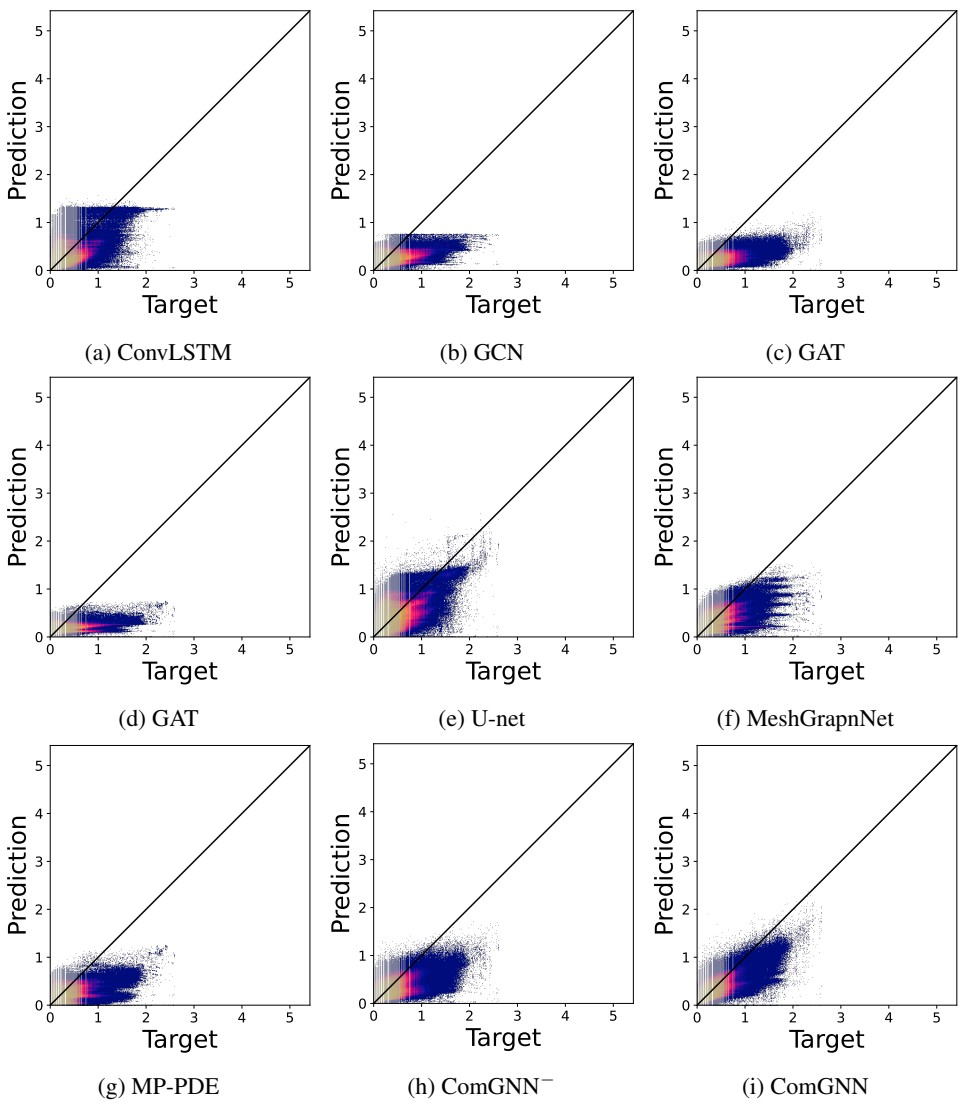

Figure 6: Scatter Plots at lead time 20 in log-log scale

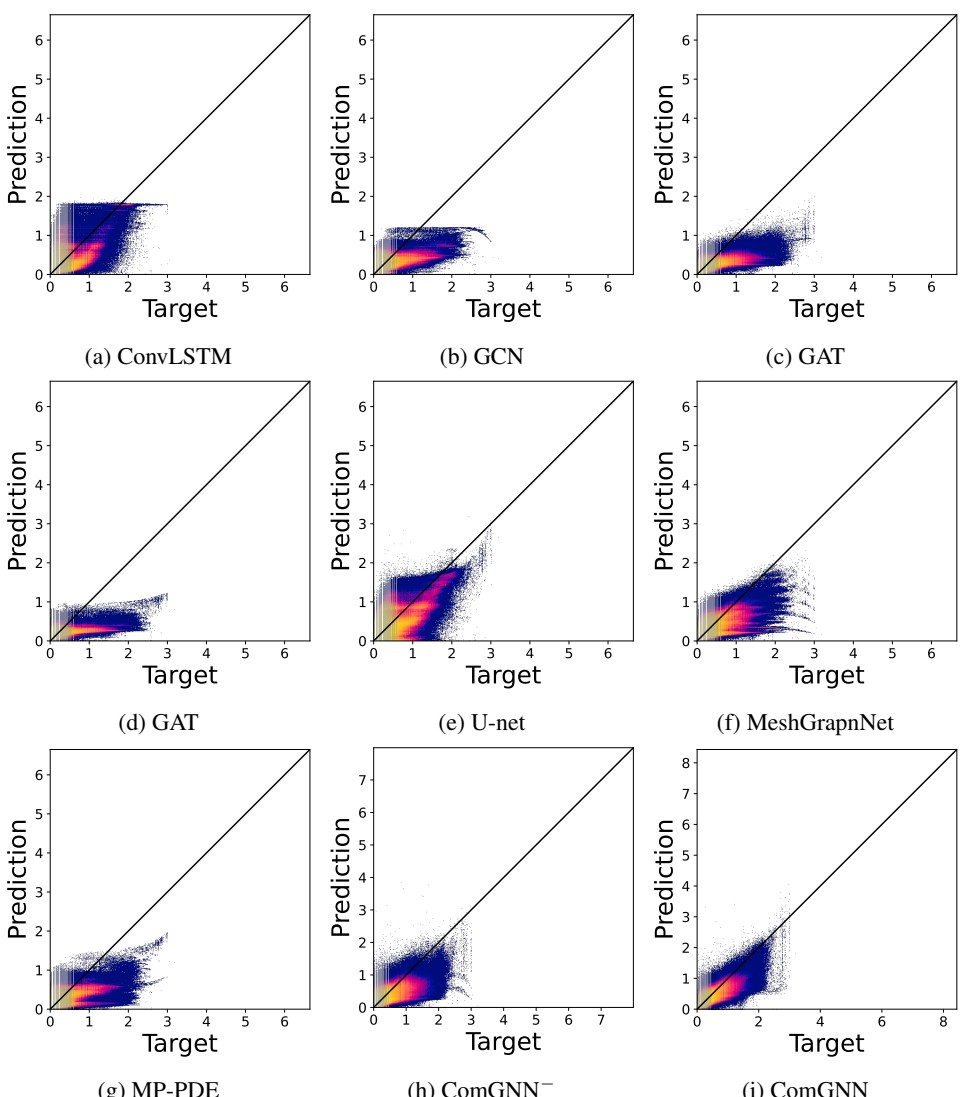

Figure 7: Scatter Plots at lead time 30 in log-log scale

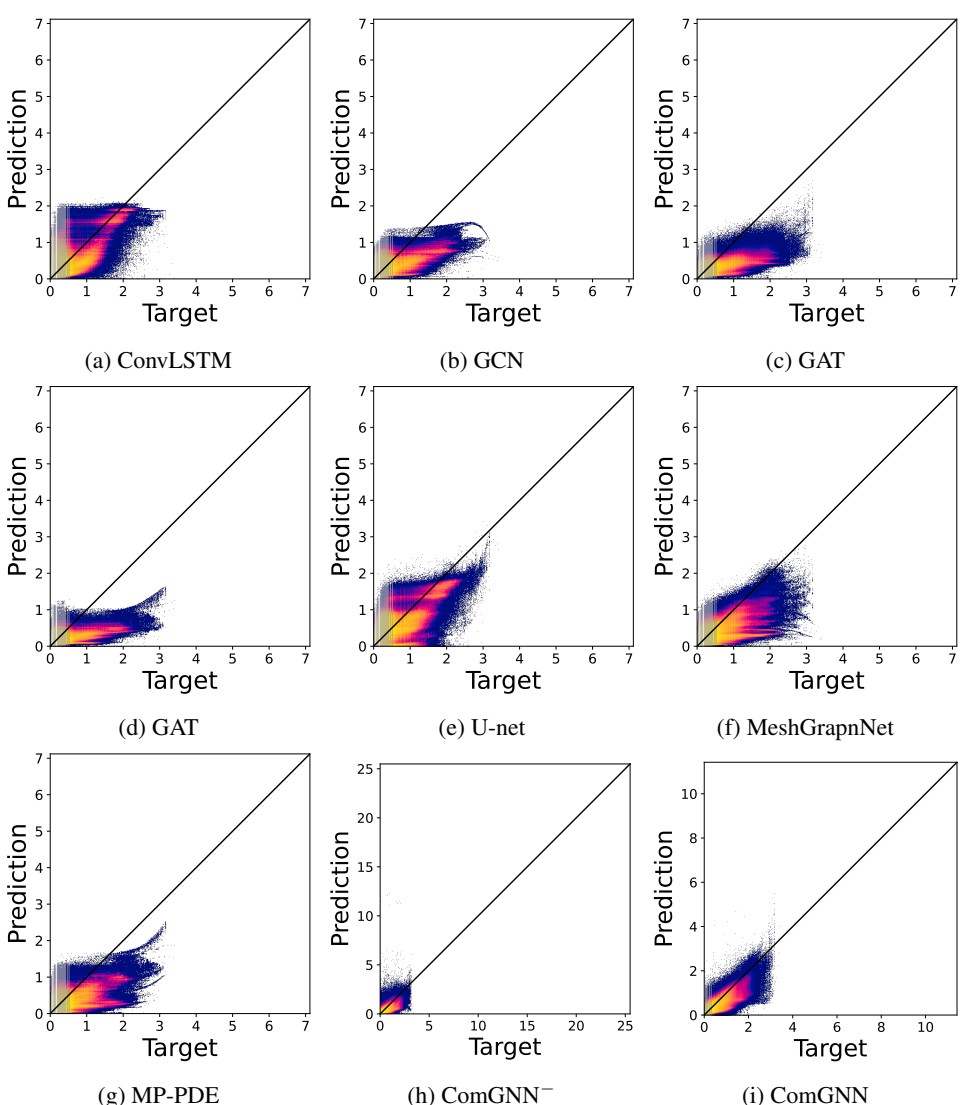

Figure 8: Scatter Plots at lead time 40 in log-log scale

