# OpenReview forum: "FLOOD SIMULATION WITH PHYSICS-INFORMED MESSAGE PASSING"
_ICLR.cc/2024/Conference — Submitted to ICLR 2024_

### Official Review · Reviewer_j8ny · 2023-10-18

**Soundness:** 2 fair
**Presentation:** 3 good
**Contribution:** 3 good
**Rating:** 5
**Confidence:** 4

**Summary:**

This paper discusses a Graph Neural Network based model to simulate the progress of a flood over a heterogenous region due to a rainfall event. The Digital Elevation Map (DEM) of the region is provided as an input. The model is based on the laws analogous to conservation of mass and momentum to govern the flow of water from one region to another, depending on the height and amount of water already accumulated. It is shown that this model can simulate the spreading of flood more accurately than other ML-based models, especially in the early stages of the flood.

**Strengths:**

The strength of the paper lies in its application - flood simulation is an important and difficult task, and the Earth System Modeling community is looking for ML-based solutions for it. This work indeed adds to it, with its interpretable, physics-based approach of representing water retention and flow as a function of elevation, along the lines of shallow water equation which forms the backbone of many hydrological models.

**Weaknesses:**

The weakness of the work, at the current stage, seem to be lying in the following:
1) The GNN-based model, including the message-passing mechanism, is not described very thoroughly.
2) The results show that CommGNN-, a variant of the final proposed model, is not able to outperform the baseline models comprehensively. The difference between CommGNN- and CommGNN is in the water retention capacity. Does it then indicate that it is this aspect, and not the model itself, which has most impact? If that is the case, then shouldn't we be trying to incorporate the water retention aspect in the existing architectures rather than proposing a new architecture?
3) We have no results on how the proposed approach compares with the hydrological/hydrodynamic models used by domain scientists for flood simulation.

**Questions:**

1) Fig 1 shows how water from two adjoining nodes at higher elevation can flow into one node at lower elevation. But how can water from a higher region divide itself into two lower locations?
2) Is elevation the only factor that influences the dispersion of water? What can be other factors and can they be taken into account in the proposed model?
3) Do the competing models take DEM or any other auxilliary information as input?
4) Is it possible to have an experimental result that shows the water retention and dispersion stages separately?
5) Can the retention abilities be different in different regions? Which factor in the model takes the retention capacity into account?
6) In the experiment, has it been assumed that each region is of uniform elevation? What is the spatial resolution of the DEM, rainfall and flood water depth map?

---

> ### Author Response · Authors · 2023-11-23
>
> We thank the reviewer for the comments. We were able to address each one of them and hope that the reviewer would consider upgrading the rating based on our responses
>
> 1) **The GNN-based model, including the message-passing mechanism, is not described very thoroughly.**
> We have rewritten the Method section providing a detailed description of our architecture.
>
> 2) **Does it then indicate that it is this aspect, and not the model itself, which has most impact? If that is the case, then shouldn't we be trying to incorporate the water retention aspect in the existing architectures rather than proposing a new architecture?**
> The design of the message passing plays a key role in the dispersion phase. The GCN and GAT models considered in our work are implemented the same way as CommGNN with the retention component but with the message-passing of the vanilla GCN and GAT,  respectively. We show that they do not perform as good as CommGNN because their message-passing is not as expressive as the one we propose following the conservation of momentum and mass
>
> 3) **We have no results on how the proposed approach compares with the hydrological/hydrodynamic models used by domain scientists for flood simulation**
> The training data is generated using the hydrodynamical model LISFLOOD-FP which we use as the ground truth. There is not enough spatio-temporal data readily available to train an ML model. Therefore, ML models are trained on the simulation output of hydrodynamic models so that they can match their accuracy.
>
> 4)**Fig 1 shows how water from two adjoining nodes at higher elevation can flow into one node at lower elevation. But how can water from a higher region divide itself into two lower locations?**
> We apply the D8 flow direction graph, which is represented by linking each node to its steepest neighbor. Thus, for each node, there is only one outgoing edge (Page 4 & Appendix A.3). The D8 flow direction has long been used by domain experts in hydrology especially when dealing with very shallow water. However, we recognize that this becomes a limitation for later stages (which is shown in our experiments in Table 2). In future work, we will investigate a better graph representation such as a dynamic graph which will be determined based on the current surface elevation and potential energy surface
>
> 5) **Is elevation the only factor that influences the dispersion of water? What can be other factors and can they be taken into account in the proposed model?**
> Soil infiltration and friction can be other factors that can be taken into account. However, these data are not readily available, especially solid infiltration extraction which is an active area of research. The incorporation of these features will result in a more involved shallow water equations and therefore will require some modification to the message passing proposed in our current work.
>
> 6) **Do the competing models take DEM or any other auxilliary information as input?**
> We use the same input, that is, DEM and precipitation, for all the models for a fair comparison.
>
> 7) **Is it possible to have an experimental result that shows the water retention and dispersion stages separately?**
> CommGNN$^-$ considers only the water dispersion. MLP, used as one of the baselines,  does not have propagation. It can be considered as a retention-only method
>
> 8) **Can the retention abilities be different in different regions? Which factor in the model takes the retention capacity into account?**
> Yes, water retention can be different over a region because the precipitation data is spatially distributed, resulting in different amounts of rainwater falling in different places of a region. Other local attributes could also be incorporated as inputs in the retention phase.
>
> 9) **In the experiment, has it been assumed that each region is of uniform elevation? What is the spatial resolution of the DEM, rainfall and flood water depth map?**
> No, they are not uniform. We are using real-world data representing watersheds with various topographies. Details about the regions are shown in Appendix A.4, Table 5, and Figure 6. The resolution of each DEM grid cell is 30m by 30m, and so are the rainfall and water depth maps.

---

### Official Review · Reviewer_g1kJ · 2023-10-26

**Soundness:** 2 fair
**Presentation:** 4 excellent
**Contribution:** 2 fair
**Rating:** 5
**Confidence:** 4

**Summary:**

The authors propose a graph neural network (GNN) model for forecasting the spatio-temporal
evolution of flooding in a geographical region based on the region's height profile and registered
precipitation data. To facilitate modeling, the region is discretized into sub-regions, each
represented as a graph node. The GNN comprises multiple sequential message-passing (MP) layers
designed to process and propagate local information, drawing inspiration from the physical
principles of mass and momentum transport as described by the shallow water equations (SWE). In
addition to this, two other physical inductive biases are considered: (i) the orientation of graph
edges between nodes follows the steepest descent direction, with only one outgoing edge per
node/sub-region, and (ii) a retention layer is applied before each MP layer to account for water
accumulation at each sub-region due to local precipitation. The model produces temporal rollouts by
recursive evaluation.

This model was trained and tested using empirical precipitation data and corresponding simulated
flooding data. The performance of the model was compared against that of an MLP, a U-Net, two
popular GNNs, and a baseline GNN previously proposed for flood modeling. The proposed model
outperforms the baselines, particularly in early-stage flood simulations.

**Strengths:**

1. The paper is well written, the concepts are clearly explained, and the figures are helpful.

2. The retention layer and the proposed MP aim to mimic the underlying physics without over-
constraining the model. While the general idea is similar to Bentivoglio et al. (2023), the key
differences are explained.

3. The model has been compared a wide range of models, including a baseline GNN model
recently proposed for flood modelling.

**Weaknesses:**

1. The reason for using the proposed GNN instead of a conventional numerical solver are not
clear. It seems not to be the goal to improve the accuracy, since the model is trained with simulated
flooding data. Is it accelerating the simulation? Runtime comparisons with numerical solvers are
not included in the paper.

2. Solely from a deep-learning viewpoint, the contributions are not significant. Another "species"
of message passing is added to an already vast range of options.

3. The term "physics-informed" used in the title is not well chosen in my opinion. This may lead to confusion
with Physics-Inspired Neural Networks (PINNs), where the governing equations are included
in the loss function. Instead, something along the lines of "physical inductive-bias" would be more suitable.

4. The use of a single out-going edge per node assumes that the information is only propagated
downhill. This may not be true for later-stage flood simulations. The authors should verify
and indicate if this inductive bias is actually helpful for test case here considered. Its possible
drawbacks should be discussed. It seems like a clear limitation for other settings without such
a strongly biased transport.

5. The authors mention the previous work of Bentivoglio et al. (2023) and Kazadi et al. (2022),
however, other similar research is missing. E.g.,
* Oliveira Santos V, Costa Rocha PA, Scott J, Th JVG, Gharabaghi B. A New Graph-Based
Deep Learning Model to Predict Flooding with Validation on a Case Study on the Humber
River. Water. 2023; 15(10):1827. https://doi.org/10.3390/w15101827
* Farahmand, H., Xu, Y. & Mostafavi, A. A spatial-temporal graph deep learning model for
urban flood nowcasting leveraging heterogeneous community features. Sci Rep 13, 6768
(2023). https://doi.org/10.1038/s41598-023-32548-x
In these two articles, the flooding models account for past states as opposed to the model
proposed here. The authors should discuss if that is needed or not. It may happen that only
the current state is relevant for inferring the water depth at the next time-point due to the
deterministic nature of the problem.

6. Whilst the model is compared to other GNNs and the GNN proposed by Bentivoglio et al.
(2023), it is not compared against the Interaction Networks (Battaglia et al. 2016), which were
shown to very suitable for Lagragian and Eurlarian simulations of fluids (Pfaff et al. 2021,
Sanchez-Gonzalez et al. 2020).

**Questions:**

1. The loss term in equation (8) penalizes the prediction of negative water depths, however, is
not better to enforce the satisfaction of this condition by the use of ReLU activation after the
last layer?

2. What is the number of parameters of each model? This is a crucial factor that usually directly determines the performance.

And 2 minor points:

3. Wrong formula for the Pearson correlation coefficient, it seems to repeat NSE?

4. Other research has considered back-propagating through multiple evaluations of the model. Could this be beneficial for the flood forecasting and long-term stability?

________

Post-rebuttal:

I'd like to thank the reviewers for the detailed update.  It is good to see that it outperforms a series of popular MP algorithms in early flooding stages. It would be important, however, to make sure all models have the same number of parameter. It seems the current comparison uses significantly different sizes.

So overall, I think the submission is improving, but I believe there are still quite a few open questions: beyond a thorough baseline comparison, the runtime would be interesting, and potentially a more detailed derivation of the method or a demonstration of broader applicability. As such I will keep my score, I'm still leaning towards the rejection side.

---

> ### Author Response · Authors · 2023-11-23
>
> We thank the reviewer for the comments. We were able to address each one of them and hope that the reviewer would consider upgrading the rating based on our responses.
>
> 1) **The reason for using the proposed GNN instead of a conventional numerical solver are not clear. It seems not to be the goal to improve the accuracy, since the model is trained with simulated flooding data. Is it accelerating the simulation? Runtime comparisons with numerical solvers are not included in the paper**
> Previous works (Bates, 2022)  have shown that ML models are faster, but they do not perform as accurately as conventional models. The current focus is to produce ML models that can perform as good as conventional models, and hopefully better. The reason why we are using flood simulation data is because there is not enough true spatiotemporal flooding data that can be used for training an ML model.
>
> 2) **Solely from a deep-learning viewpoint, the contributions are not significant. Another "species" of message passing is added to an already vast range of options**
> The novelty of our method includes (i) a two-stage paradigm in the spatiotemporal simulation of flooding when accounting for precipitation, which as external sources are rarely considered in current ML simulations of dynamic systems. (ii) The design of a message passing that follows the conservation of mass and momentum
> (iii) Applications on real-world regions, which have complex topologies compared to nicely shaped manifolds considered in current ML for dynamic systems.
> Through extensive experiments, we show that our proposed method is better than the existing alternatives. We have written the Related Work and Methods sections present our work more clearly.
>
> 3) **The term "physics-informed" used in the title is not well chosen in my opinion**
> Thank you for the suggestion. We will change the title  to “Flood Simulation With Physics-guided Message Passing" when the openreview system allows it.
>
> 4) **The use of a single out-going edge per node assumes that the information is only propagated downhill. This may not be true for later-stage flood simulations**
> We agree with the reviewers. This is the main point we showed in Table 2 and pointed out this limitation to address in future work. In future work, we will investigate a better graph representation such as a dynamic graph which will be determined based on the current surface elevation and potential energy surface. That said, a single out-going edge per node seems to work well at the early stages of simulations, and it has long been used by domain experts in hydrology when the water depth is not very high. We show these results comparing the performance between the single edge out-going edge graph and when the entire grid is used as a graph in Table 4 (Appendix A.3).
>
> 5) **In these two articles, the flooding models account for past states as opposed to the model proposed here. The authors should discuss if that is needed or not**
> Thank you for suggesting these works. In our case, based on the design of the model we tried to use past history of precipitation data in our model but we did not notice any performance improvement. However, we cannot give a definitive answer to whether the past history is relevant or not, since we might need to investigate a little bit further to figure out what feature might be suitable with past history.
>
> 6) **More baselines**
> We have compared our method to MeshGraphNet from the same authors of (Battaglia et al. 2016), (Pfaff et al. 2021), and (Sanchez-Gonzalez et al. 2020). We have added two more baselines MP-PDE, and ConvLSTM (https://proceedings.neurips.cc/paper/2015/file/07563a3fe3bbe7e3ba84431ad9d055af-Paper.pdf)
> The original versions of these baselines did not perform well. When fine-tuning them and trying out different activation functions and data normalization techniques, we found that log-transformation of the data and tanh activations boosted the performance of these methods. We therefore applied the same transformation to all other methods (including ours) for fair comparison and noticed an improvement, the results of the current version of the paper are much better than the previous ones for all the methods. Our proposed method still performs the best compared to these additional baselines
>
> 7) **Use ReLU activation after the last layer?**
> This was our initial implementation, but we found that this was a little bit difficult to train, probably due to zero gradients, the “dying ReLU” phenomenon.
>
> 8) **What is the number of parameters of each model?**
>  We have added the parameters of each model in Table 7 Appendix (A.5)
>
> 9) **Other research has considered back-propagating through multiple evaluations of the model. Could this be beneficial for the flood forecasting and long-term stability?**
> We are not sure we understand. However, this looks like an interesting idea. We would be grateful if the reviewer could share some references.

---

### Official Review · Reviewer_iyD6 · 2023-10-31

**Soundness:** 3 good
**Presentation:** 3 good
**Contribution:** 2 fair
**Rating:** 5
**Confidence:** 4

**Summary:**

The authors identify the challenges of current deep learning models in scientific applications, i.e., flood simulation here. Currently these DL models are inaccurate at modeling the early stages of flooding events and do not incorporate physical knowledge from numerical methods. The authors propose using a physics-inspired GNN to predict water depths autoregressively. Their method, ComGNN, similar to MeshGraphNet's message-passing framework is inspired by the conservation of mass and momentum in the shallow-water equations (simplification of Navier-Stokes). The proposed method is tested on real-world data covering 9 regions and 7 historical precipitation events and the results show that the model outperforms the baslines and is better at early stage flooding detection.

**Strengths:**

- It is nice that the authors try to incorporate physical information from PDEs, in this case, conservation of mass and momentum from the shallow water equations into the MeshGraphNet model.
- Nice physical motivated problem for deep learning models to improve early flood detection.
- The background on PDEs including relation between Shallow-water equations and Navier Stokes is nice.
- Nice overview of how the discretization of the PDE that the model is trained on impacts the accuracy and how CNN or pure DL models are not guaranteed to respect physical laws.
- Interesting use of directed graphs.
- Incorporating conservation of mass and momentum from the shallow water equations is critical.
- Experiments on real-world dataset LISFLOOD-FP, version 8 (Shaw et al., 2021)

**Weaknesses:**

- This method is highly related to MeshGraphNets (Pfaff et. al, "Learning Mesh-Based Simulation with Graph Networks", ICLR 2021) and needs to be compared to but adding physical information to it is beneficial.
- Numerical solutions of the shallow water equations should be added as an additional baseline.
- It should be made a bit clearer in the second paragraph of the introduction that numerical methods are still state-of-the-art in terms of accuracy compared to DL models.
- Overview of Neural Operator methods is missing from the introduction and related works.
- The first paragraph of related works reads more like a list of the works rather than describing the limitations and why further work is needed.
- When introducing GNNs in the related work section, the authors mention the GNNs can handle irregular graph data whereas image-based CNNs assume a regular grid, it would be nice to make the connection between mesh from the numerical discretization and the graph more explicit since these methods incorporate the spatial connectivity information from the mesh.
- This would be another nice reference on the connections between GNNs and finite element methods to add to the related work section ( F. Alet, A. K. Jeewajee, M. B. Villalonga, A. Rodriguez, T. Lozano-Perez, and L. Kaelbling. Graph element networks: adaptive, structured computation and memory. In International Conference on Machine Learning, pages 212–222. PMLR, 2019.).
- Also on the connection between the directed graph Laplacian operator and finite difference methods for the linear advection equation (Maddix et. al, "Modeling Advection on Directed Graphs using Matérn Gaussian Processes for Traffic Flow" (https://arxiv.org/pdf/2201.00001.pdf).
- Use of the first order Forward Euler method for the time-stepping scheme even though it has challenges to stay numerically stable (Krishnapriyan et. al, "Learning continuous models for continuous physics", 2022) and more advanced schemes such as RK4 may perform better. The underlying numerical scheme used within the DL model and truncation error analysis is quite important. Please also see Onken et. al, "Discretize-Optimize vs. Optimize-Discretize for Time-Series Regression and Continuous Normalizing Flows", 2020.
- "Since ∆t, ∆x, ∆y, and g remain constant during the entire simulation, they are simply multiplicative factors and therefore can be set to 1" - this is a major limitation. For methods such as Forward Euler to converge there is a CFL condition which bounds ∆t by ∆x and 1 is too large.
- The loss function also seems hard-coded to
- Generalization of the method - seems quite specific for the shallow-water equations and flood detection but how can the method be effectively extended to more broader PDEs?
- Application of method but no theoretical or convergence properties
- The baselines are extremely limited and compared to a CNN-based U-net method from 2015 rather than the recent state-of-the-art methods in SciML, e.g., MeshGraphNets (which is essential to compare to since I don't see how the proposed method ComGNN differs other than possibly in the loss function), MP-PDE, DINO, PINNs and Neural Operators.
- My main concern is the novelty since the method is quite similar to MeshGraphNets (Pfaff et. al, ICLR 2021) just applied on the shallow water equations.

Minor
- Start Section 3 with an overview section before going to 3.1.
- Punctuation in Eqn (1) should not have a period before it and should have a comma after it and similarly all equations should have punctuation following them.
- The references start before some of the results and a lot of the plots can be moved to an appendix to make room for more main text.

**Questions:**

- Were ablation studies conducted on why GNN architectures were chosen?
- How can this method generalize outside of flood detection and to arbitrary PDEs?
- Is there a reason why GNNs have not been explored for flood simulations? In most of the MeshGraphNet papers, they are solving Navier-Stokes equations and shallow water equations, so it isn't clear to me why it cannot be directly applied here. See also the DINO method (https://arxiv.org/pdf/2209.14855.pdf, ICLR 2023) that solves the 3D shallow water equations and compares to state-of-the-art MP-PDE Johannes Brandstetter, Daniel E. Worrall, and Max Welling. "Message passing neural PDE solvers". In
International Conference on Learning Representations, 2022 and Neural Operator methods
- Why was the forward Euler and Verlet schemes used? Were there ablation studies done on the type of numerical discretizations used, which can cause discretization errors that are propagated into the training data.
- How are the values or ∆t, ∆x, ∆y chosen and why are they all set to 1? This is far too coarse of a spatial and temporal grid. Numerical choices like this are important for stability and convergence and cannot be chosen without convergence testing.
- Why is there a mix of L1 and L2 losses in the loss function? Is this just empirically observed as stated in Appendix A.4?
- What makes ComGNN "physics-inspired" and differ from training on shallow-water simulated data with MeshGraphNets (Pfaff et. al, ICLR, 2021)?

---

> ### Author Response · Authors · 2023-11-23
>
> We thank the reviewer for the comments. We were able to address each one of them and hope that the reviewer would consider upgrading the rating based on our responses.
>
> 1) **For methods such as Forward Euler to converge there is a CFL condition which bounds ∆t by ∆x and 1 is too large.**
> We agree with the reviewer. This was not clear  in the paper. Data is generated from numerical hydrodynamic model LISFLOOD-FP,  an optimized and specialized model developed by domain experts. At fixed intervals, it outputs snapshots of water depths. However, internally it uses an adaptive time-stepping to ensure convergence. $\Delta t$ here refers to the output time (which remains the same for the entire dataset), not the internal time step used by LISFLOOD-FP.  We set $\Delta t$=1 in the current version of our work as it doesn't change
>
> 2) **Forward Euler not numerically stable, advanced schemes such as RK4 may perform better**.
> We mentioned Foward Euler to show how  $\frac{q^t-q^{t-1}}{\Delta t}$ comes from $\frac{\partial q}{\partial t}$ in Eq 8.This is only used  for our model, true data is generated by LISFLOOD-FP with an adaptive time-stepping.
>
> 3) **Numerical solutions as additional baseline**
> Given the complexity and size of the domain, the shallow water equations for flood simulation are solved numerically with optimized numerical hydrodynamic solvers (e.g, LISFLOOD-FP)  developed by domain experts. Raw numerical solvers such as FD, FEM can break in some scenarios due to the ground topography inequalities and external boundary conditions such as precipitations.
>
> 4) **Highly related to MeshGraphNets**
> Our method is different from MeshGraphNet  (i) We are dealing with system of PDEs, in MeshGraph only 1 PDE equation is used at a time (ii) Our method has 2 stages: water retention, and water dispersion. This paradigm turns out to be useful for capturing external input such as precipitation.  MeshGraphNet does not model such phenomena (iii) We have 2 message-passings that approximate spatial derivative in Eq1&Eq2 with Eq9&Eq8, respectively (iv) The design of these message-passings captures the topography of the domain and the amount of water based on conservation of mass and momentum. MeshagraphNet simply concatenates all the features.(v) We are working on domains with very complex topographies where the continuity of the water elevation can break. MeshGraphNet has been applied only on smooth manifolds.
> We compared our method to MeshGraphNet with same features as our method and showed that our approach achieves  better results.
>
> 5) **Loss function hard-coded**
> The loss function is for training purposes only, with no theoretical derivation. We found that most methods benefited from this loss, especially MeshGraphNet, with an increase of up to 20% in performance. Whereas $L_2$ and $L_1$ would require fine-tuning the learning and decay function for each individual model, which could still perform worse than this loss function.
>
> 6) **Generalization**
> While we are looking at extending our work to other physical processes, the current version of this work is specific to shallow-water equations in the context of flood simulation. We have improved the motivation of our work in the introduction.
>
> 7) **No theoretical or convergence properties**
>  We have rewritten the method section by providing a justification for the design proposed.
>
> 8) **More baselines**
> We have added MeshgGraphNet, MP-PDE, and ConvLSTM (Shi 2015)  as baselines. However, we had trouble fine-tuning DINO, PINNs, and FourierNet on our dataset because of memory constraints.
> The original versions of MeshGraphNets and MP-PDEs did not perform well. When fine-tuning them and trying out different activation functions and data normalization techniques, we found that log-transformation of the data and tanh activations boosted the performance of these methods. We therefore applied the same transformation to all other methods (including ours) for fair comparison and noticed an improvement, the results of the current version of the paper are much better. Our proposed method still performs the best compared to these additional baselines.
>
> 9) **Reason why GNNs not been explored for flood simulations?.**
> Flood simulation poses the following challenges (i) The complexity of the ground topography, which can break the continuity of flows. Even though methods such as MeshGraphNets and DINO have simulated the Navier-Stokes, their experiments are mainly conducted on simpler and smoother manifolds which ensure the continuity of the physical process. (ii) The sparsity of the data: The data is usually very sparse, which makes it difficult to train. (iii) Swift transition of states from dry to wet or wet to dry. Assuming a discretized domain, it is easy to predict water increment on a cell that has already water in it. This explains the good performance of all the methods in Table 2. However, predicting the transition from a dry cell to a wet cell is usually harder (see Table 1).

---

### Official Review · Reviewer_rnbK · 2023-11-01

**Soundness:** 3 good
**Presentation:** 3 good
**Contribution:** 2 fair
**Rating:** 3
**Confidence:** 3

**Summary:**

The paper presents a novel graph neural network (GNN) model, named ComGNN, designed for early-stage flood simulation. The model innovatively incorporates physical laws, specifically the conservation of momentum and mass from the shallow-water equations, into its framework. It operates on real-world, spatially distributed rainfall data to predict water depth over time, addressing the shortcomings of traditional hydrodynamic models and previous ML-based methods. ComGNN stands out for its two-stage approach, which first accounts for water retention at the rainfall site and then simulates the water's propagation throughout the region. The model demonstrates superior performance over existing methods, particularly in early flood stages, which is critical for timely flood risk mitigation. It's trained on real-world data from nine regions and seven historical precipitation events, showing promising results, especially in reducing root mean square error (RMSE) when predicting water depths.

**Strengths:**

1.	Using graph neural network method to successfully operates(models) in a two-stage paradigm for early-stage flood simulation given a rainfall event
2.	A message-passing mechanism on the flow direction graph has been proposed for water propagation. The experiments show that the proposed approach outperforms current approaches
3.	It is trained directly on real-world data rather than synthetic data, which may enhance the model's applicability and generalization to real-world scenarios

**Weaknesses:**

1.	The author asserts the novelty of the ComGNN model; however, the distinctiveness of its design beyond the described pipeline remains unclear. It would be prudent for the author to provide a more detailed account of the model's architecture or to draw comparisons with existing frameworks to more effectively underscore the innovative aspects of ComGNN.
2.	Within Equations 1 to 4, the manuscript appears to describe the application of a learnable multilayer perceptron (MLP) for predicting or learning the solutions of partial differential equations in the context of the conversation process. This approach does not seem to offer substantial theoretical advancement to the field. Could the author provide a more in-depth explanation of the theoretical underpinnings to clarify the contribution?
3.	The message passing scheme, a prominent feature in the title, does not convey extensive information. Would the author kindly provide relevant references for this scheme? It appears to resemble standard data flow in GNN architectures. Additionally, a precise initial definition of 'message' along with a more comprehensive explanation would be beneficial for clarity.
4.	Too much experimental figures in page 8 and 9 which makes main contents too short, author could move them in to appendix and leave more room for theoretical analysis.
5.	The current formulation of the problem may be overly simplistic for readers not well-versed in the field. It would be advisable to include the mathematical model within this section to enhance comprehension.

**Questions:**

In summary, the paper offers ComGNN, a graph neural network that simulates early-stage flood progression by accounting for rainfall retention and water propagation based on conservation principles. The authors claim that their method shows superior performance in comparison to existing approaches, especially for early flood stages, validated through empirical results on historical flood data. However,  it put too much contents in application and exrimanetal explanation and results but overlook the theoretical part, which is not suitable for ICLR. Moverover, the novelty of ComGNN didn’t revealed sufficiently in the method part.

---

> ### Author Response · Authors · 2023-11-23
>
> We thank the reviewer for the comments. We were able to address each one of them and hope that the reviewer would consider upgrading the rating based on our responses.
>
> 1) **Novelty of the ComGNN model**.
> We have expanded the method section by providing a detailed description of the model. In the revised version of the manuscript,  we show that our method operates in two stages. Each cell first retains water from the rain and then propagates it following Eq1 & Eq2  Propagation starts with the conservation of momentum (Eq2) where we use a message passing based on the topology of the ground elevation and the amount of retained water (of each cell/node) as an approximation of the spatial derivative to obtain Eq8. This produces a flow that is used in the conservation of mass (Eq1) approximated by Eq10 to output the final water depth.
> Therefore, the novelty of our method includes (i) a two-stage paradigm in the spatiotemporal simulation of flooding when accounting for precipitation data, which as external sources are rarely considered in current ML simulations of dynamic systems. (ii) The design of a message passing that follows the conservation of mass and momentum
> (iii) Applications on real-world regions, which have complex topologies compared to simpler manifolds considered in current ML for dynamic systems
>
> 2) **Eq 1 to 4 appears to describe the application of learnable multilayer**
> We have expanded the method section by showing that Eq 8 & 10 (in the current version) are the building blocks of a message passing architecture inspired by the conservation of mass and momentum.
>
> 3) **Figures in page 8 and 9 could move them in to appendix**
> Thank you for the suggestion, we have moved most figures to the appendix and have expanded the method section.
>
> 4) **The current formulation of the problem may be overly simplistic for readers not well-versed in the field. It would be advisable to include the mathematical model within this section to enhance comprehension**
> Thank you for the suggestion. We have included a description of the physics behind a flooding process in section 3 as follows. "The theoretical framework for flood modeling is based on fluid mechanics described by the 3D Navier-Stokes equation. In practice, however, the characteristic vertical length scale of the flow is very small with respect to the characteristic horizontal length scale, resulting in a constant horizontal velocity field throughout the depth of the fluid. The dynamics of a flooding process are, therefore, derived by depth integrating the 3D Navier-Stokes equation, leading to a system of non-linear PDEs called shallow-water equations..."

---

### Meta-Review · Area_Chair_rYEw · 2023-12-05

**Metareview:**

This paper presents a physics-informed graph neural network for flood simulation. The strengths of this paper include studying an important application and incorporating physical constraints into graph neural network models to outperform existing ML methods especially during the early stage of flooding. However, the paper has much room to improve its clarity, leading to quite a few clarification questions from the reviewers. Although the authors tried to answer them and revise the paper accordingly, the many piecemeal changes to the paper cannot preserve the coherence of the presentation. The paper needs a major revision to rewrite it thoroughly by presenting the motivation clearly and putting it appropriately in the context of other works. Moreover, since conservation of mass and momentum serves as a physical constraint to guide the learning process (with no guarantee though), it would help to assess how well they can be conserved when compared with other models, as well as the implications. In summary, while this work has potential to be published, major revision is needed to make it ready for publication in a top conference such as ICLR.

**Justification For Why Not Higher Score:**

Even the revised paper is below the acceptance standard.

**Justification For Why Not Lower Score:**

N/A

---

### Decision · Program_Chairs · 2024-01-16

Reject